# Simplex tensor network renormalization group for boundary theory of 3+1D symTFT

Kaixin Ji[1*], Lin Chen[2], Li-Ping Yang[3] and Ling-Yan Hung[4,5†]

**1** State Key Laboratory of Surface Physics, Department of Physics and Center for Field Theory and Particle Physics, Fudan University, Shanghai 200433, China
**2** School of Physics and Optoelectronics, South China University of Technology, 510641 Guangzhou, China
**3** Department of Physics and Chongqing Key Laboratory for Strongly Coupled Physics, Chongqing University, Chongqing 401331, China
**4** Yau Mathematical Sciences Center, Tsinghua University, Beijing 100084, China
**5** Beijing Institute of Mathematical Sciences and Applications, Beijing 101408, China

* kxji21@m.fudan.edu.cn
† lyhung@tsinghua.edu.cn

December 12, 2024

## Abstract

**Following the construction in [1], we develop a symmetry-preserving renormalization group (RG) flow for 3D symmetric theories. These theories are expressed as boundary conditions of a symmetry topological field theory (symTFT), which in our case is a 3+1D Dijkgraaf-Witten (DW) topological theory in the bulk. The boundary is geometrically organized into tetrahedra and represented as a tensor network, which we refer to as the "simplex tensor network" state. Each simplex tensor is assigned indices corresponding to its vertices, edges, and faces. We propose a numerical algorithm to implement RG flows for these boundary conditions, and explicitly demonstrate its application to a $\mathbb{Z}_2$ symmetric theory. By linearly interpolating between three topological fixed-point boundaries, we map the phase transitions characterized by local and non-local order parameters, which respectively detects the breaking of a 0-form and a 2-form symmetry. This formalism is readily extendable to other discrete symmetry groups and, in principle, can be generalized to describe 3D symmetric topological orders.**

# 1  Introduction

Generalised symmetries are now recognised to play an extremely important role in classifying and understanding phases of matter [2–8]. A universal holographic principle connects a phase with symmetries with a topological field theory in d+1 dimensions, now known as the symTFT [9–12]. By understanding boundary conditions of the symTFT, one can classify and construct symmetric theories of a given generalised symmetries. This program has been particularly successful in constructing symmetric gapped (or topological) theories, which is achieved by understanding topological boundary conditions of the symTFT [13–16].

More recently, attention has been shifted towards studying boundary conditions corresponding to symmetric gapless theories, including symmetric conformal field theories (CFT). This is a much harder problem. Explicit constructions of gapless boundaries of the symTFT are often obtained by working with well-known gapless lattice models, that can be reinterpreted or extended [1,16–21]. In the case of 1+1D CFTs, there is an important breakthrough in constructing explicit boundaries of symTFT that recovers rational CFTs [1,21]. The success depends on looking for RG fixed point that explicitly preserves symmetries. To be concrete, the symTFT considered is constructed explicitly from Turaev-Viro/Levin-Wen wavefunctions or their higher dimensional generalisations, such as the Dijkgraaf-Witten (DW) models. An RG operator that maps the boundary triangulation of a topological quantum field theory (TQFT) from a fine-grained one to a coarse grained one can be constructed from the TQFT data. Gapless boundaries are fixed points of such symmetric RG flows characterising phase transitions between the topological fixed points, the latter of which have been studied in detail when discussing symmetric gapped phases.

These RG flows in 2+1D symTFT had been studied in detail in [1], and a novel RG operator following 3+1D symTFT was explicitly constructed. There is however less progress in solving analytically for gapless fixed points for the 3+1D RG operator. Rather than pursuing an analytical solution, in this paper, we develop numerical techniques to approximate these fixed points.

In the 2+1D case, the boundary is a string-net ground state wavefunction. It is expressed as a tensor network state named the projected entangled-pair state (PEPS) [16,22–24]. In the 3+1D case, the boundary manifold is triangulated into tetrahedra. We generalize the idea of PEPS to 3D boundary by assigning a tensor to each tetrahedron. Each tensor has fourteen legs:

four legs correspond to the physical degrees of freedom at the vertices of the tetrahedron, while the remaining legs represent the entanglement through the four faces and six edges of each tetrahedron. After summing the products of all the tetrahedron tensors over all face and edge configurations, we obtain a tensor network state that lives on the vertex degrees of freedom. This design is specifically crafted to handle the entanglement among multiple tensors, which will be detailed in section 2. We will detail the technicalities we develop specially to deal with this situation whose complexity far exceeds the 2d case in section 3.

Our basic strategy is similar in spirit to that pursued in 2+1D. One constructs a boundary state that is an interpolation between two topological boundaries. The boundary is generically not a fixed point of the RG operator. Then we repeatedly apply the RG operator on the boundary state. For given interpolation parameter(s) parameterizing the initial boundary condition, repeated application of the RG operator would take it to a topological fixed point. As the interpolation parameter(s) are changed, one could reach a phase transition point. When the interpolation parameter is increased further the boundary condition would be taken to different topological fixed point under RG. This is very similar to tensor network renormalisation techniques [25–27], except that this is now carried out in our 3d tensor network in a symmetry preserving manner.

In the 2+1D case, we detect the phase transitions, by reading off the boundary condition under RG flow. However, it is basis dependent and difficult to implement in the 3+1 D case. One could also construct local and non-local operators to detect the breaking of the (generalized) symmetry [28, 29]. Their expectation values serve as order parameters for phase transitions. We construct such order parameters by insertions of topological fixed point tensors, and further develop the numerical algorithms to calculate their expectation values in the 3+1D symTFT partition function. Our methodology is tested explicitly on the 3+1D toric code model where we make use of the above method to obtain a phase diagram of 2+1 D $\mathbb{Z}_2$ phases as the boundary condition of the 3+1 D theory is changed. The methods developed in the current paper should pave the way to search of more 3d CFTs using symmetric RG flows.

Our paper is organized as follows. In section 3, we introduce the setup of our 3+1D symTFT, describing the construction of the bulk states based on DW theories and the boundary states using simplex tensor networks. In section 4, we detail the generation of the RG flow through re-triangulations, equating the partition functions of different triangulations and deriving solutions for the boundary tensors. Numerical algorithms for truncating new tensors are also introduced, with further details provided in Appendix A. In section 5, We provide explicit examples in the case of a $\mathbb{Z}_2$ bulk theory. We solve for different topological boundaries and interpolate between them. Local and non-local order parameters are construct to distinguish each phase. Their expectation values are calculated using the RG program to map the phase transitions. We also present an alternative interpretation of the simplex tensor network in the appendix B.

## 2  Construction of 2+1D partition functions from 3+1D symTFT via strange correlators

In this section, we describe how 2+1D partition functions are constructed by assigning appropriate boundary conditions to 3+1D symTFT. Our construction follows the "strange correlator" approach first considered in [14, 16], and its higher-dimensional generalization pursued in [1]. In these constructions, the symTFT adopts discrete formulations. In the case where the symTFT is 3-dimensional, the class of TQFTs considered are the so-called Turaev-Viro/Levin-Wen models [30, 31]. In higher dimensions, the Dijkgraaf-Witten models is one of the most important classes of discrete TQFTs [32], which are essentially topological gauge theories with

a discrete gauge group $G$ and explicitly formulated for arbitrary dimensions. For clarity, we focus here on symTFT that are given by 4-dimensional DW models. We will first review how the strange correlator is constructed from the 3+1D DW model with appropriate 3D boundary conditions [1].

## 2.1 Quick review on Dijkgraaf-Witten model

We begin with a brief review of the DW model construction [32]. For a more detailed review, refer to Appendix D in [33] or Section 3 in [34].

The input data for a DW model consists of an $n$-dimensional manifold $M$, a gauge group $G$, and an element of the group cohomology $H^n(G, U(1))$. The manifold $M$ is triangulated into $n$-simplices $\Delta_n$, and a branching structure is assigned to the triangulation. This is done by first giving a global ordering to the $N_v$ vertices, by numbering them from $0, 1, \cdots, N_v - 1$. Each edge thus acquires an orientation, pointing from the vertex with a smaller label to the vertex with the larger label. A group element $g_i \in G$ is assigned to each edge. This is often referred to as coloring the edges. Each simplex is assigned a chirality $\epsilon = \pm 1$ determined by their branching structure. This is illustrated for 3-simplices and 4-simplices in Figure 1.

For a group $G$, we define a $n$-cochain $\alpha(\Delta_n)$ associated to an $n$-simplex. It is a function from the edge colors of a simplex to $U(1)$. There is a condition on $\alpha$ that it vanishes unless, for every triangular face of the simplex, the edge colors satisfy the relation $g_{e_{v_0 v_1}} \times g_{e_{v_1 v_2}} = g_{e_{v_0 v_3}}$, where $e_{v_a v_b}$ denotes the edge connecting between two vertices $v_a$ and $v_b$, and the ordering of the vertices are such that $v_0 < v_1 < v_2$. This is the famous "no-flux" condition. Among admissible configurations of the edge coloring on a $n$-simplex, exactly $n$ edge colorings are independent.

To efficiently express the admissible edge colorings, we can alternatively assign a group element to each of the $n + 1$ vertices of the $n$-simplex. The edge degree of freedom is related to the vertex degree of freedom by

$$g_{e_{v_a v_b}} = g_{v_a} g_{v_b}^{-1}, \qquad \text{for } v_a < v_b. \tag{1}$$

One can readily check that this ensures that the no-flux condition is trivially satisfied around every face of the simplex. This is the discrete analogue of the pure gauge relation $\mathcal{A} = d\chi$ for gauge field $\mathcal{A}$ and pure gauge degree of freedom $\chi$ in a continuous field theory.

In a manifold with non-contractible cycle, there exist non-trivial edge colorings that cannot be solved in terms of vertex labels using equation (1) (see figure 2). However, in the current paper, the operators we consider involve only contractible regions. In such cases, the edge degrees of freedom can be freely exchanged for those on vertices. Consequently, our numerical computation in the rest of our paper will mainly be working with vertex degrees of freedom.

As a result, we can denote $\alpha$ as

$$\alpha(\Delta_n) = \alpha(g_{v_0} \cdots g_{v_n}), \qquad v_i \in \Delta_n. \tag{2}$$

There is a redundancy in this notation, since if we multiply every vertex $v_a$ element $g_{v_a}$ by the same group element, it does not change the edge configurations given by relation (1). Therefore, there is a redundancy in $\alpha$, the cochain $\alpha$ is invariant under the transformation

$$\alpha(g_0, g_1, \cdots, g_n) = \alpha(g g_0, g g_1, \cdots, g g_n), \forall g \in G. \tag{3}$$

For convenience, we sometimes shorthand $\alpha(g_0, \cdots g_n)$ by its simplex as $\alpha(\Delta_n)$.

The group cohomology $H^n(G, U(1))$ contains all unique n-cochains that further satisfies the co-cycle condition

$$\prod_{i=0}^{n+1} \alpha^{(-1)^i}(g_0, \cdots, g_{i-1}, g_{i+1}, \cdots, g_{n+1}) = 1 \tag{4}$$

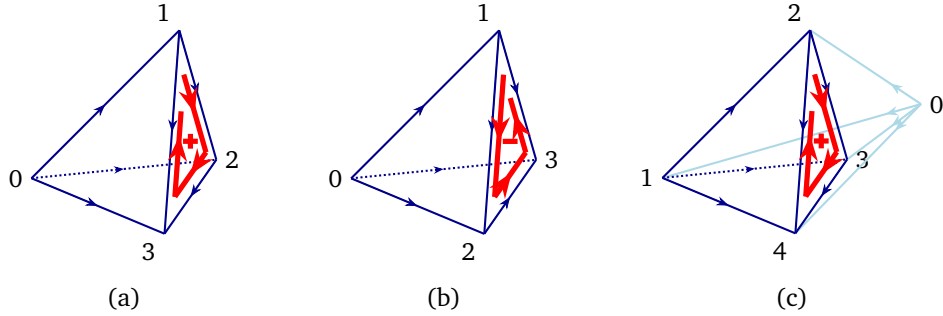

Figure 1: **Determination of the chirality ($\epsilon$) of a simplex.** For the 3-simplex in (a) and (b), we observe the loop of 123 from vertex 0. For the 4-simplex in (c), we observe the loop 234 from vertex 1. All arrows point from smaller to larger vertex labels. If the arrows of the observed loop are arranged counterclockwise (clockwise), the chirality is +1 ($-$1). For any general $n$ simplex, we observe the largest three vertexes from the fourth largest vertex.

for any array of $g_0, \cdots, g_{n+1} \in G$.

To define a DW theory, we select a cohomology class $\alpha \in H^n(G, U(1))$, which assigns $U(1)$ phase factors to each $n$-simplex $\Delta_n$. The DW partition function for $M$ is obtained as follows. First one chooses a triangulation of $M$ with $N_v$ vertices, and assigns a global branching structure to the triangulation as described above. For every given coloring of the vertices, the $n$-cochain $\alpha(\Delta_n)$ is evaluated for each $n$-simplex. The partition function on $M$ is then given by

$$Z = \frac{1}{|G|^{N_v}} \sum_{\{g_i\}} \prod_{\langle \Delta_n \rangle} \alpha^\epsilon(\Delta_n), \tag{5}$$

where the sum is over all group elements assigned to the vertices of $M$, and $|G|$ is the number of elements in group $G$. The index $\epsilon$ is the chirality for each individual simplex as defined in figure (1) . The cocycle condition (4) ensures that the partition function over a closed manifold is a topological invariant independent of the triangulation, as every triangulation can be transformed into another using the cocycle condition [32, 33].

For an open manifold $M$, the vertices lying at the boundary $\partial M$ are not summed over. Consequently, the path integral produces a function $Z(\{g_{v_i \in \partial M}\})$ that depends on the boundary vertex degrees of freedom. It is well known that this function is nothing but the ground state wave-function of the topological theory defined on $\partial M$:

$$|\Psi(\Sigma = \partial M)\rangle = \sum_{\{g_{v \in \partial M}\}} Z(\{g_{v_i \in \partial M}\})|\{g_{v \in \partial M}\}\rangle. \tag{6}$$

## 2.2 The strange correlator construction of the SymTFT partition function

As noted in the introduction, there is a holographic principle connecting a symmetric theory in $d + 1$-dimensions to a TQFT in $d + 2$ dimensions, the latter being referred to as the SymTFT in the literature. Specifically, the path-integral of a $d + 1$ dimensional symmetric theory over a manifold $\Sigma^{d+1}$ can be explicitly constructed as a $d + 2$ dimensional path-integral of a topological field theory over the manifold $\Sigma^{d+1} \times I$ for some interval $I$, with appropriate boundary condition. This is in fact the so-called "strange-correlator" proposed and then explored in [1, 14, 16].

For a theory with a discrete symmetry group $G$, the $d + 2$ dimensional SymTFT is given by the DW model. The path integral of the symmetric model is expressed as

$$Z_{d+1} = \langle \Omega(\Sigma^{d+1})|\Psi(\Sigma^{d+1})\rangle, \tag{7}$$

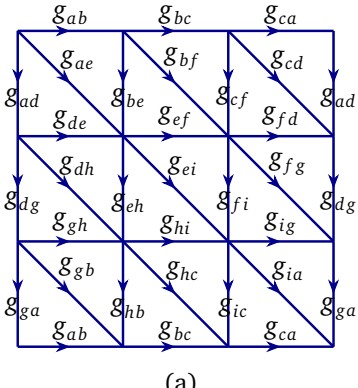 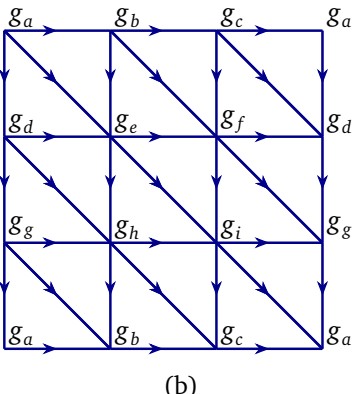

(a)                  (b)

Figure 2: **The edge and vertex coloring on a torus.** Figure (a)/(b) shows an edge/vertex coloring on a lattice triangulation of a torus. The "no-flux" condition automatically satisfies if we take equation 1 in fig(a) in terms of fig (b). However, in a non-contractible loop, say $g_{ab} \rightarrow g_{bc} \rightarrow g_{ca}$, equation 1 requires $g_{ab}g_{bc}g_{ca}$ to be the identity. For $g_{ab}g_{bc}g_{ca}$ not equal identity, the corresponding vertex coloring does not exist. In practice, these sectors are locally the same and undistinguishable for local operators. For non-local operators that wrap a non-contractible cycle, we should violate the periodicity at the "corner" element $g_a$ to sum over all possible edge coloring.

where $|\Psi(\Sigma^{d+1})\rangle$ is the wavefunction defined in (6), and $|\Psi(\Sigma^{d+1})\rangle$ is some state defined also on $\Sigma^{d+1}$, to be clarified later in this section.

In this paper, we focus on symmetric theories in 2+1 dimensions, and correspondingly consider DW theories on an open 4 dimensional manifold $M = \Sigma^3 \times I$, where $\Sigma^3$ is the 3d boundary manifold, and $I$ is an interval. Most of our results apply for generic $\Sigma^3$. In section 4, we specialize our calculations to the case $\Sigma^3 = \mathbb{T}^3$, the 3-torus.

Since we are constructing 2+1 dimensional theories as boundaries of a 3+1 dimensional DW theory, the boundary $\partial M$ is three dimensional, and inherits a triangulation into 3-simplex $\Delta_3$ from the triangulation of the 3+1 dimensional $M$. Therefore, the bra state $\langle \Omega |$ that defines the boundary condition on $\partial M$ is naturally defined as a wavefunction over a given triangulation of $\partial M$. Different triangulations can be related to each other via the cocycle condition, so we can choose a convenient triangulation of $\partial M = \Sigma^3$. A natural choice is to decompose $\Sigma^3$ into 3D cubes, which is possible at least locally. Each cube consists of several tetrahedra, as shown in figure 3. For concreteness, we consider $\Sigma^3 = \mathbb{T}^3$ when global properties are needed. It is triangulated as repeating figure 3b along three directions with periodic boundary conditions. For a given triangulation of $\partial M$, there is a canonical way to extend it to a triangulation in 4D, as shown in [1].

This extension proceeds as follows: we add an extra vertex $S$ located in the interior of $M$, but not on $\partial M$. The extended 4D triangulation is obtained by connecting this new vertex to every vertex on $\partial M$. Thus, each tetrahedron is extended to a 4-simplex. All new edges are internal to $M$. To avoid ambiguity, we assume all such edges are pointing from $S$ to the boundary vertices. The path-integral of the DW theory over $M$, constructed from $\partial M$, is given by

$$\Psi(\{g_{v \in \partial M}\}) = \frac{1}{|G|} \sum_{g_S} \prod_{\langle \Delta_4 \rangle} \alpha^{\epsilon}(\Delta_4). \tag{8}$$

It describes a ground state $|\Psi\rangle$ of the DW theory on $\partial M$. In the above, the degrees of freedom of the internal edge inside $M$ should be summed over for a given boundary coloring, and this

is achieved by summing over the degree of freedom of the bulk vertex.

## 2.3 The boundary state as a simplex tensor network

With this preparation, we are now ready to write down an ansatz for the wavefunction $\Omega(\{g_{v\in\partial M}\})$ that defines $\langle\Omega|$ in (7). The key idea is that the symmetric 2+1 D models we consider are *local*. Therefore, we expect that $\Omega(\{g_{v\in\partial M}\})$ should respect locality of $\partial M$. A natural way to encode the locality of the wavefunction is to express it as a tensor network, where the wavefunction is a product of tensors associated with each tetrahedron $\Delta_3$ on $\partial M$. These tensors carry auxiliary degrees of freedom that are contracted locally with those of neighboring tetrahedra. These tensors should also carry dangling legs corresponding to the degrees of freedom on the vertices belonging to $\partial M$. In this sense, the $\langle\Omega|$ should be understood as a higher dimensional generalization of the PEPS of a string net model that has previously been used to construct 2+1 D strange correlators [16, 24].

Our generalization differ from the string net PEPS in two key ways. First, as a higher dimensional state, the tensors are geometrically 3 dimensional objects. And we includes indices corresponding to vertices, edges and faces of 3-simplices to express the entanglement between these objects. Second, in the string net PEPS, some corner index shared by more than 2 tensors are represented using a double index structure, which is expanded into a loop [16]. While this double index structure requires more computational memory, it facilitates bond truncations in 2D PEPS. In 3D, the vertex and edge legs are shared among multiple tensors. The double index trick would be impractical due to its high computational cost, a concern also noted [35]. Therefore, we use a single index for the vertex or edge legs, as the index $p$, $q$ or $s$ in the explicit summation diagram (11). This change is necessary to keep the computation tenable with reasonable resources. We refer to this new approach as the **simplex tensor network**, emphasizing its 3D nature based on 3-simplices, drawing an analogy to the 2D "projected entangled simplex states", which were also developed to handle multi-vertex entanglement [36, 37].

The explicit ansatz is as follows. We assign a tensor to each $\Delta_3$ with vertices $ijkl$ by

$$T_{ijkl} \equiv T^{e_{ij},e_{ik},e_{il},e_{jk},e_{jl},e_{kl}}_{f_{jkl},f_{ikl},f_{ijl},f_{ijk}}(g_i, g_j, g_k, g_l), \tag{9}$$

or shorthand $T^E_F(V)$, where $e$ (or $E$), $f$ (or $F$) and $g$ (or $V$) represent respectively the edge, face and vertex elements of the simplex tensor. We will take the edge and face indices as auxiliary indices to be contracted between neighboring tensors sharing the same edge/face to build a wave function depending on the vertex degrees of freedom on the boundary $\partial M = \Sigma^3$, i.e,

$$\Omega(\{g_{v\in\partial M}\}) = \sum_{\{E\},\{F\}} \prod_{\langle\Delta_3\rangle} T^E_F(V). \tag{10}$$

We impose the requirement that given the branching structure of $M$ and inherited in $\partial M$, simplices of the same chirality should be assigned the same tensor $T$. For simplices of opposite chirality, we assign tensors $\bar{T}$ to them.

Different states $\langle\Omega|$ may not necessarily describe different phases. The actual phase of the model can be identified by considering renormalization group flow of the partition function [1]

$$Z = \sum_{\{g_{v\in\partial M}\in G\}} \langle\Omega|\Psi\rangle. \tag{11}$$

The summation is over all configurations of all the boundary vertices. The RG flow of the 3d model can be translated into flows of $\langle\Omega|$, and in turn the simplex tensors $T^E_F(V)$. It is one

---

[1]From a topological point of view, there should be a renormalization factor $\frac{1}{|G|}$ for each summation of $g_i$. Because we are mostly interested in the expectation values rather than the actual partition function, from now on we omit the factor for simplicity.

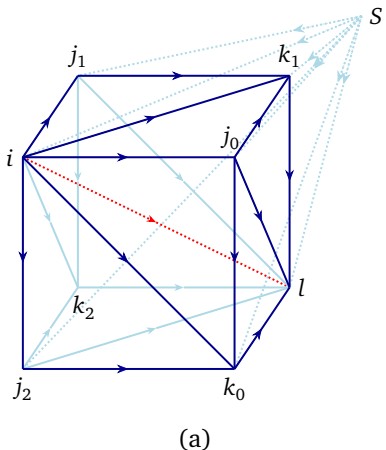 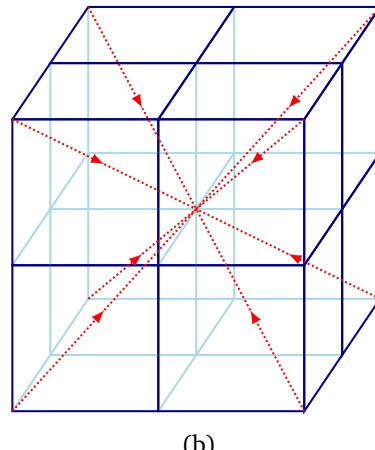

| (a) | (b) |

Figure 3: **The triangulation of the manifold** $M$**.** Figure (a) shows a small cube composed of six boundary $\Delta_3$s, with vertices $ijkl$. A bulk vertex $S$ is connected to each vertex, forming six bulk $\Delta_4$s with vertices $Sijkl$. Figure (b) shows one unit cell composed of 8 small cubes as described in (a), with all diagonal pointing to the center of (b).For simplicity, the vertex $S$ is omitted in figure (b). This triangulation is specifically designed to preserve its structure under successive RG steps.

major goal in this paper to explore numerical methods to implement RG flows on $T_F^E(V)$ that explicitly preserve the group symmetry of the 2+1 dimensional model with the help of the 3+1 dimensional DW theory.

## 3 RG procedure

In this section, we provide a step-by-step explanation of how to generate the RG flow of $T_F^E(V)$ by systematically changing the triangulation of the boundary of the four dimensional manifold.

For a given triangulation, each tetrahedron $\Delta_3$ on the boundary contributes a tensor $T_{ijkl}$ or $\bar{T}_{ijkl}$, and each 4-simplex $\Delta_4$ in the bulk contributes $\alpha_{ijklm}^{\epsilon} \equiv \alpha^{\epsilon}(g_i, g_j, g_k, g_l, g_m)$. Here, $T_{ijkl}$ and $\bar{T}_{ijkl}$ are shorthand notations for the tensors. Explicit equations with tensor indices are provided in Appendix A.

### 3.1 Boundary re-triangulation and equations of invariance

First, we illustrate how the boundary triangulation can be modified. Consider two neighbouring tetrahedra $\Delta_3$ (0234 and 1234) on the boundary, as shown in figure 4a. A bulk vertex $S$ connects to these tetrahedra, forming two bulk 4-simplices $\Delta_4$ ($S0234$ and $S1234$) whose boundaries are 0234 and 1234 respectively. According to the previously introduced rules, the configuration in figure 4a contributes the term

$$\sum_{f_{234}} T_{0234} \bar{T}_{1234} \alpha_{S0234} \alpha_{S1234}^{-1}, \tag{12}$$

where $f_{234}$ is the face index to be contracted, as this face is shared by 0234 and 1234.

The triangulation in Figure 4a can be modified to that in Figure 4b, where the boundary is re-triangulated into three $\Delta_3$ (0123, 0124 and 0134). The bulk is then triangulated into three $\Delta_4$ ($S0123$, $S0124$ and $S0134$). This new configuration in figure 4b contributes

$$\sum_{e_{01},f_{012},f_{013},f_{014}} T_{0123}' \bar{T}_{0124}' T_{0134}' \alpha_{S0123} \alpha_{S0124}^{-1} \alpha_{S0134}, \tag{13}$$

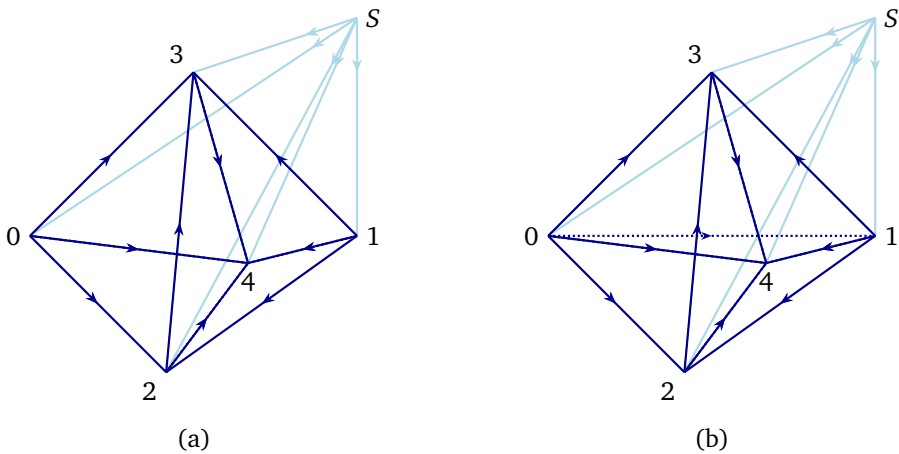

Figure 4: **Re-triangulation of two simplices.** At the boundary, 0234 and 1234 is re-triangulated into 0123, 0134, 0124. This process is carried out in two steps. First, $S0234$ and $S0134$ are re-triangulated into $S0123$, $S0134$, $S0124$, and $01234$ using the co-cycle condition. Then, $01234$ is combined with 0234 and 1234 to obtain 0123, 0134, and 0124. In figure (b), the edge 01 passes through the 234 plane. The transformation of the simplex tensors is expressed in equation (16).

where $e_{01}$ is the internal edge index to be contracted, and $f_{012}, f_{013}, f_{014}$ are the face indices to be contracted.

The partition function must remain invariant under different triangulations. Thus, the tensors $T$ and $T'$ are related by

$$\sum_{f_{234}} T_{0234}\bar{T}_{1234}\alpha_{S0234}\alpha_{S1234}^{-1} = \sum_{e_{01},f_{012},f_{013},f_{014}} T'_{0123}\bar{T}'_{0124}T'_{0134}\alpha_{S0123}\alpha_{S0124}^{-1}\alpha_{S0134}, \tag{14}$$

Using the co-cycle condition for $g_S, g_0, g_1, g_2, g_3, g_4$,

$$\alpha_{01234}\alpha_{S1234}^{-1}\alpha_{S0234}\alpha_{S0134}^{-1}\alpha_{S0124}\alpha_{S0123}^{-1} = 1, \tag{15}$$

we obtain

$$\sum_{f_{234}} \alpha_{01234} T_{0234}\bar{T}_{1234} = \sum_{e_{01},f_{012},f_{013},f_{014}} T'_{0134}\bar{T}'_{0124}T'_{0123}. \tag{16}$$

When the tensor $T$ depends only on vertex indices (i.e. the edge and face bond dimensions are 1), a re-triangulation invariant tensor $T$ can be obtained by solving

$$\alpha_{01234} T_{0234}\bar{T}_{1234} = T_{0134}\bar{T}_{0124}T_{0123}. \tag{17}$$

We denote the solution as $\beta$, which is a 3-cochain, and require $\bar{\beta}$ to be $\beta^{-1}$. The re-triangulation invariance condition then becomes

$$\alpha_{01234} = \prod_{i=0}^{4} \beta^{(-1)^i}(g_0, \cdots, g_{i-1}, g_{i+1}, \cdots, g_4). \tag{18}$$

This equation is precisely the Frobenius condition for a topological theory. The solutions $\beta$ thereby construct topological fixed-point boundaries.

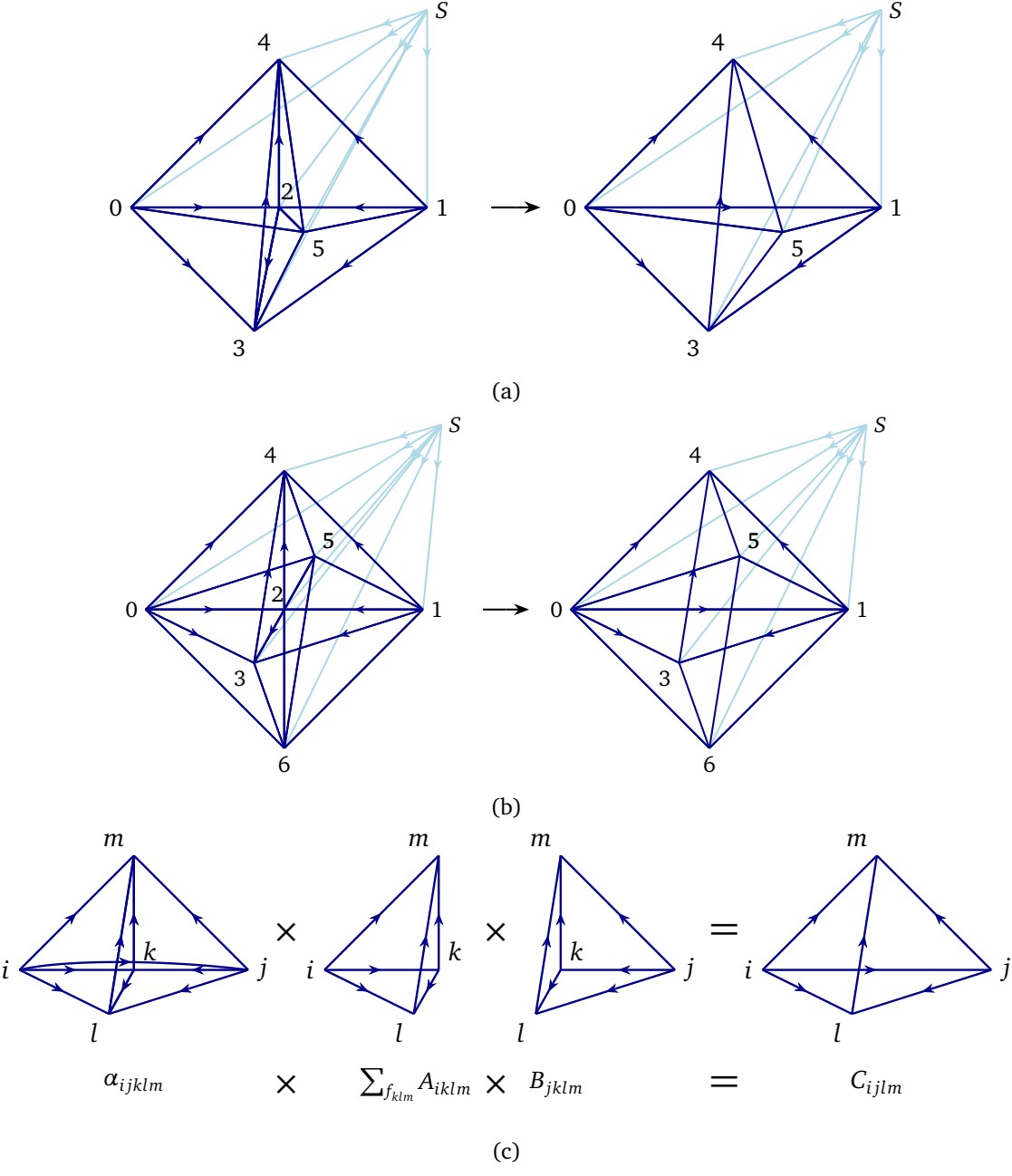

Figure 5: **Re-triangulation of pairs of simplices.** In figure (a), three pairs of $\Delta_4$ surrounding vertex 2 are re-triangulated to three larger $\Delta_4$. The re-triangulation process for the simplex tensors is described by equation (23). The solution for the new tensor on the right-hand side is given by equation (25). In figure (b), the re-triangulation of four pairs of simplices is illustrated. For any such re-triangulation, the solution can be expressed in equation (24) and is visually represented in (c).

## 3.2 Removal of boundary vertex by re-triangulations

By suitable re-triangulations, we can even eliminate a boundary vertex. In figure 5a, we show that the boundary vertex 2 can be removed by re-triangulation. On the left hand side of figure 5a, there are six boundary $\Delta_3$s $(0234, 1234, 0245, 1245, 0235, 1235)$ and their corresponding bulk $\Delta_4$ counterparts surrounding vertex 2. Thus, the contribution from the left-hand side is

$$\sum_{f_{023},f_{123},f_{024},f_{124},f_{025},f_{125},f_{234},f_{235},f_{245}}^{g_2,e_{02},e_{12},e_{23},e_{24},e_{25}} \alpha_{S0234}\alpha_{S1234}^{-1}\alpha_{S0245}\alpha_{S1245}^{-1}\alpha_{S0235}^{-1}\alpha_{S1235}T_{0234}\bar{T}_{1234}T_{0245}\bar{T}_{1245}\bar{T}_{0235}T_{1235}. \tag{19}$$

For another triangulation as shown on the right hand side of figure 5a, the vertex 2 is removed. Now there are three boundary $\Delta_3$s $(0134, 0145, 0135)$ and their corresponding bulk $\Delta_4$ counterparts, which are obtained by combining $02ij$ and $12ij$ $(ij = 34, 45, 35)$. The contribution from the right-hand side is

$$\sum_{f_{013},f_{014},f_{015}}^{e_{01}} \alpha_{S0134}\alpha_{S0145}\alpha_{S0135}^{-1}T'_{0134}T'_{0145}\bar{T}'_{0135}. \tag{20}$$

The partition function is required to be invariant under different triangulations, i.e.

$$\sum_{f_{023},f_{123},f_{024},f_{124},f_{025},f_{125},f_{234},f_{235},f_{245}}^{g_2,e_{02},e_{12},e_{23},e_{24},e_{25}} \alpha_{S0234}\alpha_{S1234}^{-1}\alpha_{S0245}\alpha_{S1245}^{-1}\alpha_{S0235}^{-1}\alpha_{S1235}T_{0234}\bar{T}_{1234}T_{0245}\bar{T}_{1245}\bar{T}_{0235}T_{1235}$$

$$= \sum_{f_{013},f_{014},f_{015}}^{e_{01}} \alpha_{S0134}\alpha_{S0145}\alpha_{S0135}^{-1}T'_{0134}T'_{0145}\bar{T}'_{0135}. \tag{21}$$

Using the co-cycle condition

$$\alpha_{012ij}\alpha_{S12ij}^{-1}\alpha_{S02ij}\alpha_{S01ij}^{-1}\alpha_{S012j}\alpha_{S012i}^{-1} = 1, \tag{22}$$

where $ij = 34, 45, 35$, we can obtain the equation

$$\sum_{f_{023},f_{123},f_{024},f_{124},f_{025},f_{125},f_{234},f_{235},f_{245}}^{g_2,e_{02},e_{12},e_{23},e_{24},e_{25}} \alpha_{01234}T_{0234}\bar{T}_{1234}\alpha_{01245}T_{0245}\bar{T}_{1245}\alpha_{01235}\bar{T}_{0235}T_{1235} = \sum_{f_{013},f_{014},f_{015}}^{e_{01}} T'_{0134}T'_{0145}\bar{T}'_{0135}. \tag{23}$$

Here we have also used the fact that $\alpha_{S012i}\alpha_{S012j}^{-1}$ (for $ij = 34, 45, 53$) appear on the same side and cancel each other out. This cancellation is independent of the specific index ordering, ensuring that the solution holds for any pair of simplices.

We observe that

$$T'_{01ij} = \alpha_{012ij}\sum_{f_{2ij}}T_{02ij}\bar{T}_{12ij} \tag{24}$$

satisfies equation (23) automatically when $f_{01i}$ is interpreted as the combination of $f_{02i}, f_{12i}, e_{2i}$, and $e_{01}$ is interpreted as the combination of $g_2, e_{02}, e_{12}$. That is, the tensor indices of $T'$ are formed from the combinations of those of $T$. This solution is formally written as $T' = F(\alpha, T, \bar{T})$, where we define a function $F : (\alpha, A, B) \to C$. It takes a cocycle $\alpha$ and two tensors $A, B$ to a new tensor $C$ as shown in figure 5c, and defined as

$$F(\alpha, A, B) \equiv C_{ijlm} = \alpha_{ijklm}\sum_{f_{klm}}A_{iklm}B_{jklm}\Big|_{f_{ijl}=(e_{kl},f_{ikl},f_{jkl}),f_{ijm}=(e_{km},f_{ikm},f_{jkm})}^{e_{ij}=(g_k,e_{ik},e_{jk})}. \tag{25}$$

Here, $e_{ij} = (g_k, e_{ik}, e_{jk})$ represents all possible combinations of $g_k, e_{ik}, e_{jk}$, and similarly for $f_{ijl}$ and $f_{ijm}$.

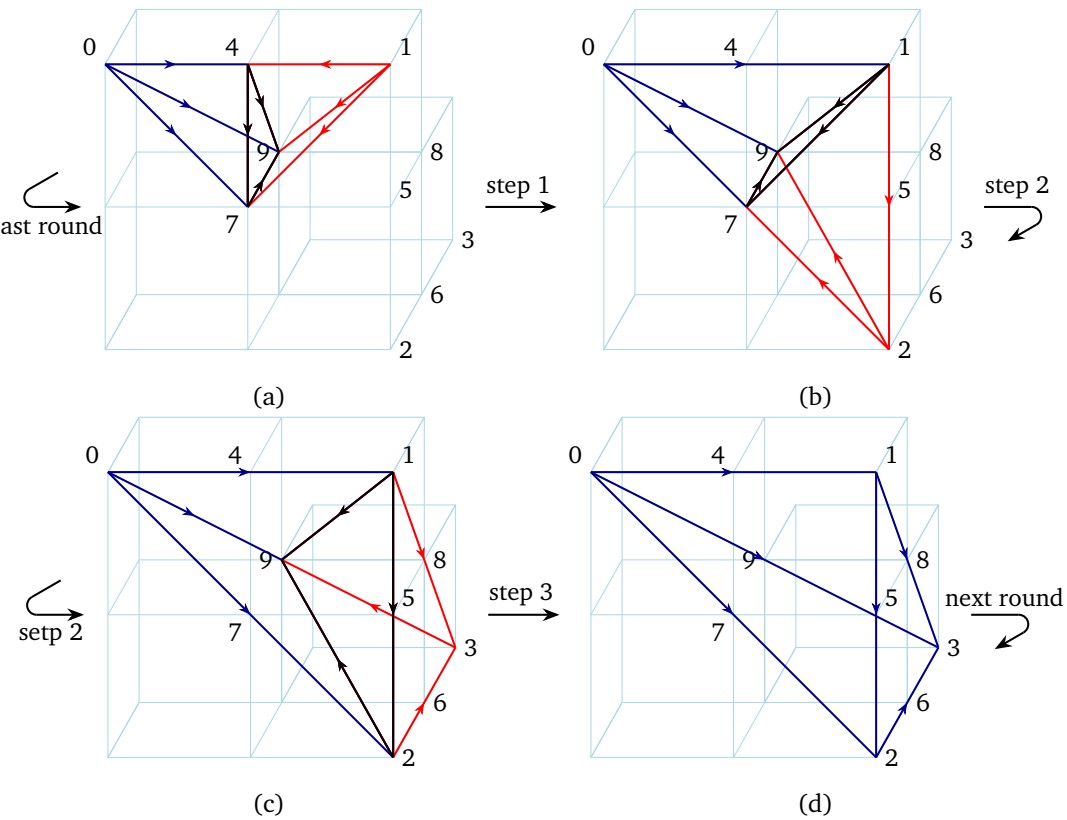

Figure 6: **The RG flow of the boundary tensors.** Step 1: Remove the edge centers in each unit cell, such as vertices 4, 5, and 6. From Fig. (a) to Fig. (b), the simplices 0479 and 1479 are combined into 0179. Step 2: Remove the face centers in each unit cell, such as vertices 7 and 8. From Fig. (b) to Fig. (c), the simplices 0179 and 1279 are combined into 0129. Step 3: Remove the body center in each unit cell, such as vertex 9 in this case. From Fig. (c) to Fig. (d), the simplices 0129 and 1239 are combined into 0123. The simplex 0123 becomes the new "0479" for the next iteration. All labels are for illustrative purposes and represent local ordering.

This map combines the tensor indices of $A, B$ are combined to be the tensor indices of $C$ without any truncation. Thus, it can be used to generate exact solutions for combining any pairs of $\Delta_3$. For example, the solution for both figures 5a and 5b can be given in the form $T' = F(\alpha, T, \bar{T})$ and $\bar{T}' = F(\alpha^{-1}, \bar{T}, T)$. These new tensors serve as the starting point for finding truncated (approximate) tensors in our numerical algorithm, as detailed below and in appendix A.

## 3.3   RG flow of boundary tensors and bond truncations

We have showed how to remove a physical vertex $i$ on the boundary $\partial M$ by suitable re-triangulation while keeping the partition function invariant. In other words, we can gener-atd $\Omega'(\cdots, g_{i-1}, g_{i+1}, \cdots)$ from $\Omega(\cdots, g_{i-1}, g_i, g_{i+1}, \cdots)$ while having $\langle \Omega' | \Psi' \rangle = \langle \Omega | \Psi \rangle$, where $\langle \Omega |, \langle \Omega' |$ are the simplex tensor network states respectively with and without the boundary vertex $i$. Similarly, $|\Psi\rangle, |\Psi'\rangle$ are the ground states of the DW theory corresponding to different triangulations.

Starting from the boundary triangulation defined in figure 3, we can now generate the RG flow for the boundary states by applying the re-triangulation procedure outlined above. One RG round consists of three steps, and combines eight boundary $\Delta_3$ into a new $\Delta'_3$, as shown

in figure 6.

In step 1, we remove all vertices at the center of each edge of the $2 \times 2 \times 2$ cube, such as 4, 5 and 6 in figure 6a. For instance, vertex 4 is removed by combining 0479 and 1479 to form 0179 as depicted from figure 6a to 6b. The new tensors $T'$, $\bar{T}'$ after the combination are given by $T' = F(\alpha^{-1}, T, \bar{T})$ and $\bar{T}' = F(\alpha, \bar{T}, T)$, where $T, \bar{T}$ are the tensors for the $\Delta_3$ in figure 6a.

In step 2, we remove all the vertices at the center of each face of the $2 \times 2 \times 2$ cube, such as 7 and 8. As an example, vertex 7 is removed by combining 0179 and 1279 to form 0129 as shown in figure 6b to 6c. The new tensors are given by $T'' = F(\alpha, T', T')$ and $\bar{T}'' = F(\alpha^{-1}, \bar{T}', \bar{T}')$.

In step 3, we remove the vertices in the center of the $2 \times 2 \times 2$ cube, such as the vertex 9. This is achieved by combining 0129 and 1239 to form 0123 as shown in figure 6c to 6d. The new tensors are given by $T''' = F(\alpha^{-1}, T'', \bar{T}'')$ and $\bar{T}''' = F(\alpha, \bar{T}'', T'')$.

After completing one RG round, the $\Delta_3$ in figure 6d will serve as the starting point for the next RG round. This process can be iterated as one requires. The RG equations with explicit indices are provided in appendix A.

Up to this point, we have derived an exact RG flow for the tensors that preserves the partition function invariance. However, the cost of this process is that the dimensions of the tensor indices continue to grow, as shown in Equation (25). In practice, and as required by the spirit of RG, we need to keep the bond dimensions (the dimensions of the tensor indices) under a fixed value. To achieve this, we must truncate the increasing bond dimensions.

Here, we briefly describe the numerical algorithms used for truncation, with further details provided in the appendix A. After the tensor combination (e.g., as in Equation (24)), two face indices and one edge index——specifically $f_{02i}, f_{12i}, e_{2i}$)——become the face index $f_{01i}$ of $T'$. Since each face is shared by only two $\Delta_3$, the contraction of the face index $f_{01i}$ involves only two tensors, which can be regarded as the usual matrix multiplication of the two tensors. Therefore, the face indices $f_{01i}$ can be truncated using techniques similar to the Higher-order tensor network renormalizational group (HOTRG) first introduced in [27].

The edge index $e_{01}$ of $T'$ is a combination of the indices $g_2, e_{02}, e_{12}$ of $T$, and should be truncated. However, the edge is generally shared by more than two $\Delta_3$, so the contraction of the edge index $e_{01}$ involves more than two tensors. To handle this, we employ a gradient descent algorithm to find the best truncated tensor, which minimizes the squared error relative to the exact tensor after contracting the indices.

In some special cases, the tensor network structure and thus the RG algorithm can be simplified. One example is, if $T_{ijkl} = f_1(g_i \cdot g_j) f_2(g_j \cdot g_k) f_3(g_k \cdot g_l)$, it can be factorized to terms associated with virtual legs. As a result, the boundary states admit a more compact tensor network representation and can be efficiently computed. This is exemplified by using HOTRG to compute the classical 3d Ising partition function [27]. The same problem can be view as computing using our formalism with tensors given in equation(27), with however less accuracy. The simplification is possible because the edge indices can be efficiently rewritten as face indices. In general, such rewriting would demand exponentially more memory resources.

# 4 Phase transition between fixed point tensors of $\mathbb{Z}_2$ fields

In this section, we explore the phase diagrams in the space of models with $\mathbb{Z}_2$ symmetries using the above formalism. It is well known that in 2+1D, symmetry protected topological (SPT) phases with symmetry group $G = \mathbb{Z}_2$ are classified by group cohomology $H^3(\mathbb{Z}_2, U(1)) = \mathbb{Z}_2$ [33, 38]. This classification implies the existence of two distinct $\mathbb{Z}_2$ symmetric phases. In addition, the $\mathbb{Z}_2$ symmetry could be spontaneously broken, leading to a "ferromagnet" phase.

It would be very interesting to explore the phase diagram in theory space that interpolates between these gapped phases using our framework. To describe phases that carry $\mathbb{Z}_2$ symmetry,

we should look for boundary conditions of 3+1D $\mathcal{Z}_2$ DW theory. The latter is classified by $H^4(\mathbb{Z}_N, U(1)) = \mathbb{Z}_1$, i.e. there is only 1 cohomology class of 4-cocycles. They are equivalent to $\alpha(\Delta_4) = 1$, wherever all the faces of the colored 4-simplex satisfies the no-flux condition described earlier, which is solved by placing degrees of freedom on the vertex that satisfies equation (1).

For this 4-cocycle, there are three sets of solutions to equation 18. There are the topological fixed-point (TFP) of the RG procedure, associated with the three gapped phases mentioned above. We will identify order parameter for each of these phases, and introduce interpolation parameters that interpolate between these boundary conditions. By plotting the order parameters against the interpolation parameter, we can trace the phase diagram of the $\mathbb{Z}_2$ phases. A phase diagram interpolating between these three phases have also been considered in [39]. Here, we explore it via numerical computations in our framework.

We emphasis our numerical setups here: we restrict the bond dimension of the edge and face indices to be no greater than 2. The vertex bond dimension, which corresponds to physical indices contracted with the ground state of the 3+1D DW model, is always 2 given the $\mathbb{Z}_2$ bulk theory. We assume that the tensors associated to 4-simplices of opposite chirality for a given branching structure to be Hermitian conjugates, i.e., $T = \bar{T}^*$ or $\beta = \bar{\beta}^*$ for equations (26)-(29). This guarantees that the 2+1D partition function, derived from the strange correlator, is real. Because we do not include any non-contractable cycles in our operators, we impose periodic boundary condition for our $|\Omega\rangle$ on $\mathbb{T}^3$, as explained in figure 2.

## 4.1 Topological fixed points and physical interpretations

We begin by writing down the TFP tensors where the face and edge bond dimensions are both equal to one. Since $\alpha(\Delta_4) = 1$, equation (18) simplifies to the 4-cocycle condition. The TFP boundary states are trivial fixed-point wave functions in 2+1D, classified by $H^3(\mathbb{Z}_2, U(1))$ [33, 40].

We use the convention that $\mathbb{Z}_2 = \{1, -1\}$, and assume the group operation is math multiplication. That is, $1 \times 1 = 1, 1 \times -1 = -1, -1 \times 1 = -1$ and $-1 \times -1 = 1$. There are three TFPs: two from $H^3(\mathbb{Z}_2, U(1)) = \{\beta_0, \beta_{-1}\}$ and one from $H^3(\mathbb{Z}_1, U(1)) = \{\beta_1\}$. $\mathbb{Z}_1$ means only the trivial subgroup of $\mathbb{Z}_2$ are non-zero components. We explicitly fix the co-boundary condition by letting

$$\beta_{\text{SPT}_0}(g_i, g_j, g_k, g_l) = 1, \tag{26a}$$

$$\beta_{\text{SPT}_1}(g_i, g_j, g_k, g_l) = \begin{cases} -1, & g_i g_j = g_j g_k = g_k g_l = -1 \\ 1, & \text{otherwise} \end{cases}, \tag{26b}$$

$$\beta_{\text{SB}}(g_i, g_j, g_k, g_l) = \begin{cases} 1, & g_i g_j = g_j g_k = g_k g_l = 1 \\ 0, & \text{otherwise} \end{cases}. \tag{26c}$$

Here, $\beta_{\text{SPT}_0}$ / $\beta_{\text{SPT}_1}$ are the trivial/twisted superposition of all configurations, which are consistent with the construction of the trivial / non-trivial 2+1D $\mathbb{Z}_2$ SPT phases [38]. They are symmetric under global $\mathbb{Z}_2$ action.

On the other hand, $\beta_{\text{SB}}$ represents the polarized state in the symmetry breaking (SB) phase. As indicated by $H^3(\mathbb{Z}_1, U(1))$, the only non-zero configuration in this phase corresponds to all $g_i$ being the same.

The symmetric and symmetry-broken phases are separated by Ising-type phase transitions. To make it manifest, we construct a continuous path between $\beta_{\text{SPT}_0}$ and $\beta_{\text{SB}}$, controlled by a parameter $J$:

$$T_{\text{Ising}}[J](g_i, g_j, g_k, g_l) = e^{(g_i \cdot g_j + 2g_j \cdot g_k + g_k \cdot g_l)J/8}. \tag{27}$$

Here $[\cdot]$ denotes the parameter of the tensor function, to distinguish it from the vertex elements. The edge $ij$ and $kl$ are shared by eight tetrahedrons, while edge $jk$ are shared by four (see figure 3a). The overall symTFT partition function is

$$\sum_{g_i} \prod_{\langle i,j \rangle} e^{Jg_i g_j}. \tag{28}$$

This is equivalent to a classical 3D Ising model with coupling strength $J$. It is straightforward to show that $\beta_{\text{SPT}_0} = T_{Ising}[J = 0]$, $\beta_{SB} \propto T_{Ising}[J \to \infty]$. The critical tensor at the phase transition should construct a 3D Ising CFT partition function. We computed the average magnetization $\langle g_i \rangle$, which is the average value for $g_i = \pm 1$. The phase transition point is located at $J = 0.22(4)$. This critical coupling strength is consistent with results obtained from other methods, including Monte Carlo simulations $J = 0.22165463$ [41] and tensor network renormalization $J = 0.221653$ [27].

## 4.2  Phase diagrams with local and non-local operators

The phase transitions between $\beta_{\text{SPT}_0}$ and $\beta_{\text{SPT}_1}$ is less well-studied. Both phases are symmetric under $\mathbb{Z}_2$, with repect to different 2-form symmetry of the SPT order. It is established that, with $\mathbb{Z}_2$ symmetry preserved, these two phases are separated by either spontaneous symmetry breaking or long-range entangled states, the latter including CFTs or intrinsic topological orders [34].

In general for the $\mathbb{Z}_2$ fields, we can assign any tensor with 16 variables. To include all three TFPs in a straightforward manner, we restrict our phase space by linearly interpolating the three tensors. We define such a tensor by parameter $x$ and $y$ as

$$T[x, y] = (1-y)\beta_{SB} + y\left(\frac{1+x}{2}\beta_{\text{SPT}_0} + \frac{1-x}{2}\beta_{\text{SPT}_1}\right) = \begin{cases} 1, & g_i g_j = g_j g_k = g_k g_l = 1 \\ xy, & g_i g_j = g_j g_k = g_k g_l = -1 \\ y, & \text{otherwise.} \end{cases} \tag{29}$$

$[\cdot]$ again denotes the parameter of tensor function. Parameter $x$ controls the interpolation between the two SPT phases. Parameter $y$ controls the interpolation between the SB phase and SPT phases.

Figure 7a plots the expectation value of the vertex elements $\langle g_i \rangle$ against the interpolation parameters $x, y$. It captures the violation of the $\mathbb{Z}_2$ symmetry. The two blue corner belong to the two SPT phases, while the yellow region corresponds to the SB phase. The absolute value of the SPT "area" is small due to our linear parametrization, which should be the exponential of usual parameters such as temperature.

For $y = 1$, $T[x, y]$ is $\mathbb{Z}_2$ symmetric. However, the $\mathbb{Z}_2$ symmetry is spontaneously broken along the path. We can further analytically continue this path by complex $x$. We fix $y = 1$ and consider complex values of $x$ and plot $\langle g_i \rangle$ in figure 7b. Near the unit circle and away from the imaginary axis, the system tends to stay in the $\mathbb{Z}_2$ symmetric phase, forming two blue crescents. However, they are still separated by spontaneous symmetry breaking phase near the imaginary axis. It numerically verifies that, any path connecting the two $\mathbb{Z}_2$ SPT phases will go through an SB phase, for boundary states given by the 3-cochains. We also conjecture that the tips of the crescents may correspond to undetermined CFT other than Ising, which are the intersections of phase transition lines.

Figure 7b is mirror symmetric about both horizontal and vertical axes. The horizontal symmetry arises from the hermitian conjugacy: the expectation values are independent of the sign

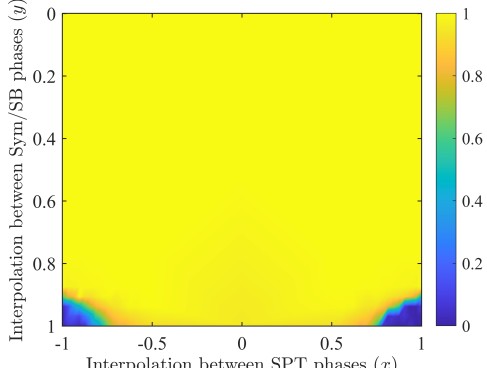

(a) $\langle \mathbf{g_i} \rangle$ **for initial tensor $\mathbf{T[x,y]}$.** The left and right blue regions represent the non-trivial and trivial $\mathbb{Z}_2$ SPT phases, respectively. The yellow region corresponds to the symmetry-broken phase.

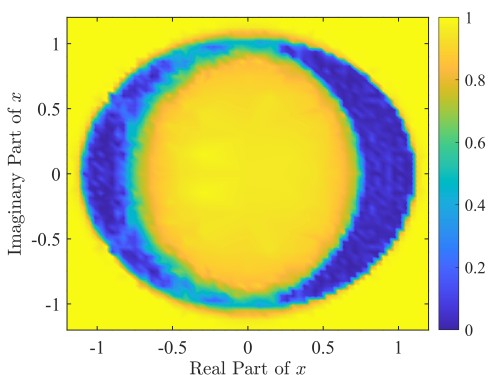

(b) $\langle \mathbf{g_i} \rangle$ **for initial tensor $\mathbf{T[x,1]}$ with complex x.** When the absolute value of $x$ is close to 1 and far from the imaginary axis, the system tends to remain in the $\mathbb{Z}_2$ symmetric phase, forming two blue crescent-shaped regions. The left and right crescents correspond to the non-trivial and trivial $\mathbb{Z}_2$ SPT phases, respectively.

Figure 7

of the imaginary part. The vertical symmetry can be explained through the duality between the left and right sides that commutes with the $\mathbb{Z}_2$. Notice that $T[-x, y] = \beta_{SPT_1} T[x, y]$. This is a local duality operation that, when applied to each 3-simplex, flips the state vertically. For this duality transformation, we define the local operator $\hat{D}$ acting on any 3-simplex $ijkl$ by

$$\hat{D}_{ijkl} = \begin{cases} -1, & g_i g_j = g_j g_k = g_k g_l = -1, \\ 1, & \text{otherwise.} \end{cases} \tag{30}$$

A similar operation is also discussed in [39].

To distinguish between the two SPT phases, we can insert the fixed-point tensors from either one of them into the boundary states. Within the corresponding phase, the insertions serve as a topological defect line that respects the higher-form symmetry, which is broken in the other phase. For concreteness, we insert the FPT from the non-trivial $SPT_1$ state and define the local operator $\hat{M}$ acting on any 3-simplex $ijkl$ (see figure 8a for the physical interpretations),

$$\hat{M}_{ijkl',ijkl} = \frac{1}{|G|} \sum_{g_l} \hat{D}_{ijll'} \hat{D}_{ijll'} \hat{D}_{ijll'}. \tag{31}$$

These operators can be interpreted as non-local membrane operators acting on the $ijk$ face. We insert different numbers of $\hat{M}$ on a boundary plane, and plot their expectation values in figure 8. It picks out the non-trivial SPT phase where it is effectively acting as an identity operator. With increasing area of the membrane operator (i.e. the number of $\hat{M}$ increases), the distinction the two SPT phases becomes more apparent. This suggests that the membrane operator is a well-behaved non-local order parameter for the SPT phase transitions, which in our case detects the breaking of the 2-form symmetries.

## 5 Conclusion

In this paper, we explored in detail how gapless symmetric phases can be systematically searched using RG flows derived from the symTFT framework [1]. We developed the RG procedure and

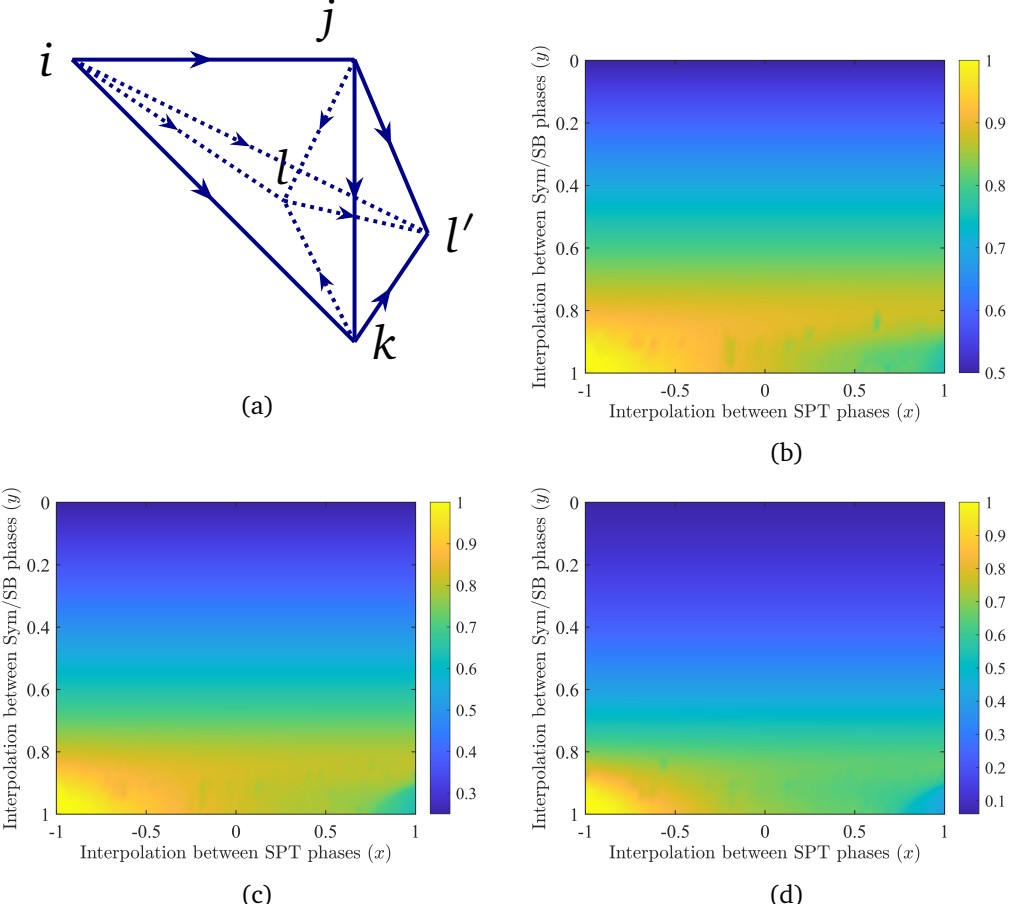

(a)

(b)

(c) (d)

Figure 8: **Definition of membrane operators and their expectation values.** Figure (a) illustrates the physical interpretation of a single membrane operator. It inserts three $SPT_1$ tensor, $ijll'$, $ikll'$, and $jkll'$, around the original simplex $ijkl$, and combines them into a new tensor $ijkl'$. Figure (b) maps the expectation value of a single $\hat{M}$ with basis state given by $T[x, y]$. Figure (c) and (d) depict the expectation values for a group of two and four $\hat{M}$ operators aligned together, respectively. As the order of $\hat{M}$ increases, the two SPT phases at the corners become more distinct.

a corresponding numerical algorithm to implement these RG flows for 3D symmetric theories as boundaries of 4D symTFTs.

Using this framework, we can distinguish between different phases by analyzing the expectation values of local and non-local order parameters. This is exemplified in the case of $\mathbb{Z}_2$ phases, where the symTFT corresponds to the 4D $\mathbb{Z}_2$ toric code model. In this context, there are three topological fixed-point tensors: one representing a symmetry-broken phase and two representing distinct symmetry-protected topological phases. We linearly interpolate between these tensors and take that as a parametrisation of the phase space, and search for phase boundaries and critical points in the phase space. The average magnetization, which is a local order parameter, effectively distinguishes the symmetry-broken phase from the SPT phases. Meanwhile, a non-local membrane parameter, constructed from topological fixed-point tensors, discriminate the two SPT phases making use of their differing 2-form symmetries.

A notable by-product of this study is the introduction of a novel 3D tensor network formalism, termed the simplex tensor network. This tool was employed and developed here to implement RG flows for symTFT partition functions. We illustrated the method implementing a $\mathbb{Z}_2$ bulk theory. Our study should lay the ground work for generlization to any finite discrete symmetry group $G$, such as $\mathbb{Z}_N$ or $\mathbb{Z}_N \times \mathbb{Z}_M$. In the current paper, we mostly introduced degrees of freedom on vertices of the graph. However, our method can be readily generalized to include face and edge degrees of freedom, which are needed in describing symmetry-enriched topological states [34]. Furthermore, the simplex tensor network may serve as a standalone representation of 3D quantum states, with potential applications in simulating 3+1D time evolution or computing variational ground state energies. These aspects remain unexplored in the current work.

This is a first step towards systematically searching for 2+1 D symmetric phases numerically by interpolating different boundary conditions given to the 3+1 D symTFTs. Several challenges, however, warrant further investigation. First, the truncation algorithm for edge bonds needs improvement. The current gradient descent algorithm is sensitive to initial guesses and hyper-parameters. It requires multiple runs to achieve near-optimal solutions in practice. This limitation is evident in figures 7 and 8, which still exhibits fluctuations in regions that should appear smooth. Second, the gauge redundancy for local tensors needs to be better understood. In 2D tensor networks, gauge transformations have significantly simplified RG algorithms [8, 42, 43]. For simplex tensor networks, edge bonds are shared among multiple tensors, leading to more complicated gauge conditions. Investigating how generalized symmetries could serve as gauge conditions for these networks is a promising direction for future exploration.

Third, recovering precise CFT data at gapless fixed points is of critical importance. Achieving this may require methodologies analogous to the fuzzy sphere approach [44]. Additionally, there is a purported tricritical point [39], and we believe that refinements in our construction could bring us closer to this point. We will return to these important questions in a future publication.

# Acknowledgements

We thank Rui-Zhen Huang, Shenghan Jiang, Laurens Lootens, Ce Shen, Antoine Tilloy, Frank Verstraete, Chenjie Wang and Wen-Tao Xu for useful discussions. LYH acknowledges the support of NSFC (Grant No. 11922502, 11875111). L. -P. Y. was supported by the National Natural Science Foundation of China, NSFC (Grants No. 11874095). L. C. acknowledges support from the NSFC Grant No. 12305080, the Guangzhou Science and Technology Project with Grant No. SL2023A04J00576, and the startup funding of South China University of Technology.

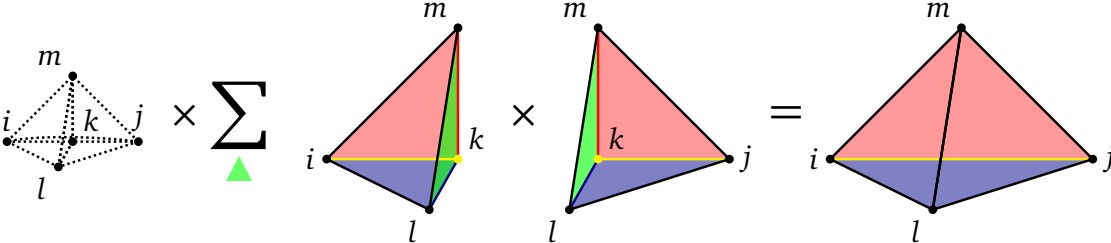

Figure 9: **An illustration of equation (32).** The edge and face legs on the left-hand side and their combinatorial legs on the right-hand side are indicated with matching colors. Yellow legs $g_k$, $e_{ik}$ and $e_{jk}$ are combined into the yellow leg $e_{ij}$. Red legs $e_{km}$, $f_{ikm}$ and $f_{jkm}$ are combined into the red leg $f_{ijl}$. Blue legs $e_{kl}$, $f_{ikl}$ and $f_{jkl}$ are combined into the blue leg $f_{ijl}$. The green leg $f_{klm}$ is summed over.

## A  Tensor algorithms

This appendix provides a systematic introduction to the algorithms used for computing the coarse-grained tensors and expectation values. In A.1, we provide explicit forms of the RG equations discussed in section 3.3 and illustrated in figure 6. This is meant to provide a direct reference for tensor calculations in our algorithm. In A.2, we explain the bond truncation procedure, with an emphasis on the loss function for the gradient descent. In A.3, we sketch on how to compute expectation values with insertions of impurity tensors.

### A.1  RG equations for tensor

The equation 24 or 25 in explicit indices as in figure 9 is given by

$$
\alpha(g_i, g_j, g_k, g_l, g_m) \times \sum_{f_{klm}} A^{e_{ik},e_{il},e_{im},e_{kl},e_{km},e_{lm}}_{f_{klm},f_{ilm},f_{ikm},f_{ikl}}(g_i, g_k, g_l, g_m) \times B^{e_{jk},e_{jl},e_{jm},e_{kl},e_{km},e_{lm}}_{f_{klm},f_{jlm},f_{jkm},f_{jkl}}(g_j, g_k, g_l, g_m)
$$

$$
= C^{e_{ij},e_{il},e_{im},e_{jl},e_{jm},e_{lm}}_{f_{jlm},f_{ilm},f_{ijm},f_{ijl}}(g_i, g_j, g_l, g_m)\Big|^{e_{ij}=(g_k,e_{ik},e_{jk})}_{f_{ijl}=(e_{kl},f_{ikl},f_{jkl}),f_{ijm}=(e_{km},f_{ikm},f_{jkm})}. \tag{32}
$$

Again, $a = (b, c, d)$ means $a$ takes all possible combinations of $b, c, d$.

We now give some example equations for each RG step. Tensors of opposite chirality are complex conjugates of each other and are denoted as $T$ and $\bar{T}$. In the first step of the RG (figure 6a), for tensor 0479 and 1479 combined into the tensor 0179,

$$
T'^{e_{01},e_{07},e_{09},e_{17},e_{19},e_{79}}_{f_{179},f_{079},f_{019},f_{017}}(g_0, g_1, g_7, g_9) = f_1(\alpha^{-1}, T, \bar{T}) \equiv \sum_{f_{479}} \Big[ \alpha^{-1}(g_0, g_1, g_4, g_7, g_9)
$$

$$
T^{e_{04},e_{07},e_{09},e_{47},e_{49},e_{79}}_{f_{479},f_{079},f_{049},f_{047}}(g_0, g_4, g_7, g_9)\bar{T}^{e_{14},e_{17},e_{19},e_{47},e_{49},e_{79}}_{f_{479},f_{179},f_{149},f_{147}}(g_1, g_4, g_7, g_9)\Big|^{e_{01}=(g_4,e_{04},e_{14})}_{f_{017}=(e_{47},f_{047},f_{147}),f_{019}=(e_{49},f_{049},f_{149})} \Big]. \tag{33}
$$

For the tensors of opposite chirality, say 1289, we have $\bar{T}' = f_1(\alpha, \bar{T}, T)$.

In step 2, for tensor 0179 and 1279 to combine into 0129,

$$
T''^{e_{01},e_{02},e_{09},e_{12},e_{19},e_{29}}_{f_{129},f_{029},f_{019},f_{012}}(g_0, g_1, g_2, g_9) = f_2(\alpha, T', \bar{T}') \equiv \sum_{f_{179}} \Big[ \alpha(g_0, g_1, g_2, g_7, g_9)
$$

$$
T'^{e_{01},e_{07},e_{09},e_{17},e_{19},e_{79}}_{f_{179},f_{079},f_{019},f_{017}}(g_0, g_1, g_7, g_9)T'^{e_{12},e_{17},e_{19},e_{27},e_{29},e_{79}}_{f_{279},f_{179},f_{129},f_{127}}(g_1, g_2, g_7, g_9)\Big|^{e_{02}=(g_7,e_{07},e_{27})}_{f_{012}=(e_{17},f_{017},f_{127}),f_{029}=(e_{79},f_{079},f_{279})} \Big]. \tag{34}
$$

For the tensors of opposite chirality, say 1239, we have $\bar{T}'' = f_2(\alpha^{-1}, \bar{T}', \bar{T}')$.

In step 3, for tensor 0129 and 1239 to combine into 0123,

$$T'''^{e_{01},e_{02},e_{03},e_{12},e_{13},e_{23}}_{f_{123},f_{023},f_{013},f_{012}}(g_0,g_1,g_2,g_3) = f_3(\alpha^{-1},T'',\bar{T}'') \equiv \sum_{f_{179}} \Big[ \alpha^{-1}(g_0,g_1,g_2,g_3,g_9)$$

$$T''^{e_{01},e_{02},e_{09},e_{12},e_{19},e_{29}}_{f_{129},f_{029},f_{019},f_{012}}(g_0,g_1,g_2,g_9) \bar{T}''^{e_{12},e_{13},e_{19},e_{23},e_{29},e_{39}}_{f_{239},f_{139},f_{129},f_{123}}(g_1,g_2,g_3,g_9)\Big|^{e_{03}=(g_9,e_{09},e_{39})}_{f_{013}=(e_{19},f_{019},f_{139}),f_{023}=(e_{29},f_{029},f_{239})} \Big].$$
(35)

For the tensors of opposite chirality, $\bar{T}''' = f_3(\alpha,\bar{T}'',T'')$.

## A.2 Bond truncation

After each RG step, there are two face legs and one edge leg to be truncated. They involve different numbers of tensors, as illustrated in figure 10. We use step 1 as an example. The other two steps are analogous.

In step 1, the composite legs are $e_{01},f_{017},f_{019}$ as shown in figure 10a. Their summation involve eight tensors. To be clear, the tetrahedron $e_1e_2v_1v_2$ in figure 10a is the same simplex as the 0179 in figure 6b.

We rearrange the indices into different groups $p,q,r,s,t$ according to their summing relations, see figure 10. $p$ and $q$ are shared by all eight tensors; $r_j$ and $s_j$ are shared between neighboring tensors; $p/r_j$ represent composite edge/face elements summed in this diagram. $q$ and $s_j$ represent vertex and edge elements not summed here. $t_j$ are the rest elements that appear only once in this diagram. By the above setting, $p = e_1e_2, q = (e_1,e_2), r_j = e_1e_2v_j, s_j = (v_j,e_1v_j,e_2v_j)$, $t_j = (v_jv_{j+1},e_1v_jv_{j+1},e_2v_jv_{j+1})$ where $j = 1,\cdots,8$ with periodicity (i.e. $j = 9$ is identified with $j = 1$). It leads to a more compact form of the tensors

$$A_{pqr_{i-1}r_is_{i-1}s_it_{i-1}} = T'^{e_1e_2,e_1v_{i-1},e_1v_i,e_2v_{i-1},e_2v_i,v_{i-1}v_i}_{e_2v_{i-1}v_i,e_1v_{i-1}v_i,e_1e_2v_i,e_1e_2v_{i-1}}(e_1,e_2,v_{i-1},v_i) \tag{36a}$$

$$\bar{A}_{pqr_{i+1}r_is_{i+1}s_it_i} = \bar{T}'^{e_1e_2,e_1v_{i+1},e_1v_i,e_2v_{i+1},e_2v_i,v_{i+1}v_i}_{e_2v_{i+1}v_i,e_1v_{i+1}v_i,e_1e_2v_i,e_1e_2v_{i+1}}(e_1,e_2,v_{i+1},v_i) \tag{36b}$$

where $i = 2,4,6,8$ with periodicity.

The goal is to find truncated tensor $B_{p'qr'_{i-1}r'_is_{i-1}s_it_{i-1}}$ and $\bar{B}_{p'qr'_{i+1}r'_is_{i+1}s_it_i}$, with bond dimension $d_{p'} < d_p, d_{r'_i} < d_{r_i}$, such that the error $|E-F|^2$ is minimized. Here, $E$ and $F$ are the products of tensors after summing the composite legs, and $|\cdot|^2$ denotes the sum of the squared tensor elements. Explicitly,

$$E_{q,s_1,\cdots,s_8,t_1,\cdots,t_8} = \sum_{p,r_1,\cdots,r_8} \prod_{i=2,4,6,8} A_{pqr_{i-1}r_is_{i-1}s_it_{i-1}} \bar{A}_{pqr_{i+1}r_is_{i+1}s_it_i} \tag{37a}$$

$$F_{q,s_1,\cdots,s_8,t_1,\cdots,t_8} = \sum_{p',r'_1,\cdots,r'_8} \prod_{i=2,4,6,8} B_{p'qr'_{i-1}r'_is_{i-1}s_it_{i-1}} \bar{B}_{p'qr'_{i+1}r'_is_{i+1}s_it_i} \tag{37b}$$

To truncate the face legs of tensor $A$ and $\bar{A}$, we perform HOSVD. The procedure begins by computing $R_{r_ir_k} = \sum_{pqr_js_ls_mt} A_{pqr_ir_js_ls_mt}A_{pqr_kr_js_ls_mt}$. Next, we perform eigenvalue decomposition of the matrix $R = U\Sigma U^\dagger$. Here $\Sigma$ is the diagonal matrix of eigenvalues, sorted in decreasing order. We truncate the $d_{r_i} \times d_{r_i}$ matrix $U$ by keeping the top $d_{r'_i}$ eigenvectors, resulting in a truncated $d_{r_i} \times d_{r'_i}$ matrix $U'$. Similarly we compute $L_{r_jr_k} = \sum_{pqr_is_ls_mt} A_{pqr_ir_js_ls_mt}A_{pqr_ir_ks_ls_mt}$ and truncate to the corresponding $d_{r_j} \times d_{r'_j}$ matrix $V'$. After truncation, the partially truncated tensor is:

$$A'_{pqr'_ir'_js_ks_lt} = \sum_{r_i,r_j} A_{pqr_ir_js_ks_lt_m}U'_{r_i,r'_i}V'_{r_j,r'_j}. \tag{38}$$

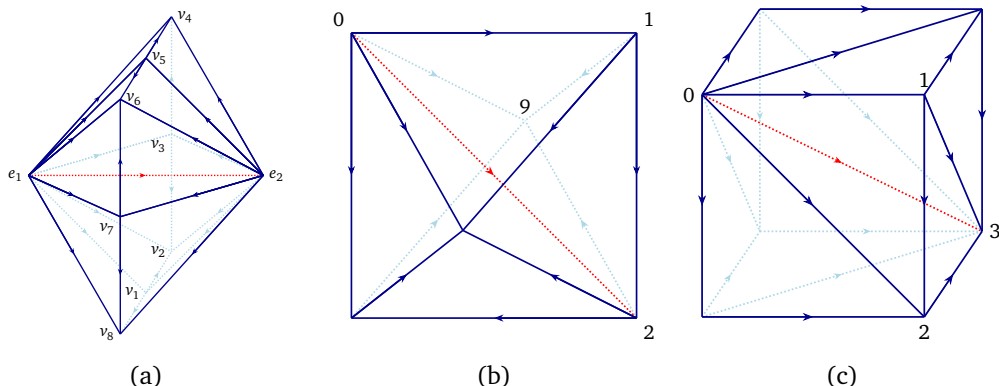

(a)           (b)           (c)

Figure 10: **The summation diagram for tensor truncation.** Figures (a), (b), and (c) show the tensors involved in steps 1, 2, and 3 of one round of the RG flow, as illustrated in Figure 6. In Figure (a), a new set of indices is used to clarify the summation required for truncation. The tetrahedron $e_1 e_2 v_1 v_2$ corresponds to the same tensor as 0179 in Figure 6b.

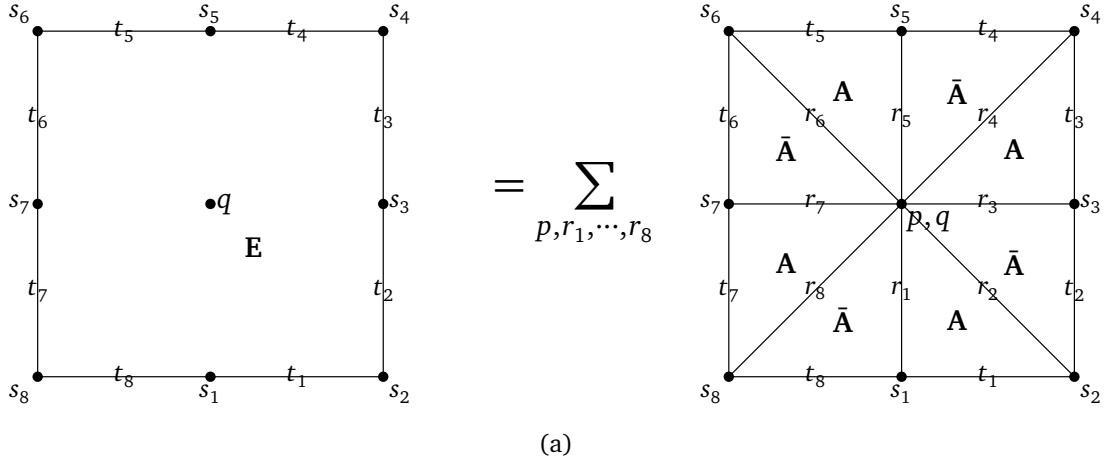

(a)

Figure 11: **A systematic illustration of equation 37a.** The left hand side is the tensor $E$. The right hand side is the sum of the product of $A$ and $\bar{A}$. Legs $p, q$ are shared between eight tensors. Legs $r, s$ are shared between two tensors. Legs $t$ belong to each individual tensor.

We use the best $d_{p'}$ components of the $A'$ along axis $p$, which has the lowest error, as the initial guess for tensor $B$. Then, we optimize $B$ using gradient descent, specifically Limited-memory Broyden–Fletcher–Goldfarb–Shanno algorithm with adaptive learning rate, to minimize $|E - F|^2$ [45].

Things are similar for step 2 and step 3. We skip the trouble of repeating this story. Please refer to figure 10b and 10c for the diagrams of summation.

### A.3  Computing expectation value with impurity tensor(s)

In this section, we reframe the established method of computing expectation values using impurity tensors [46] within our model. To compute the expectation value of a local operator $\hat{O}$, we apply it to the initial tensor $T$, generating an impurity tensor $P = \hat{O}T$. At each RG step, we get the new impurity tensor by $P' = f_i(P, T)$ while the other tensors follow the usual equation $T' = f_i(T, T)$. We denote the updated tensors as $T' = A$ and $P' = C$, following the same conventions as in the previous section. First, we perform the usual truncation on $A$ to obtain $B$ as above. Next, we truncate $C$ to obtain $D$, by minimizing the error of $|\tilde{E} - \tilde{F}|^2$. Here, $\tilde{E}$ and $\tilde{F}$ are similar to $E$ and $F$, with one tensor substituted by the impurity tensor $C$ or $D$:

$$\tilde{E}_{q,s_1\cdots,t_1\cdots} = \sum_{p,r_1\cdots} C_{pqr_1r_2s_1s_2t_1}\bar{A}_{pqr_3r_2s_3s_2t_2} \prod_{i=4,6\cdots} A_{pqr_{i-1}r_is_{i-1}s_it_{i-1}}\bar{A}_{pqr_{i+1}r_is_{i+1}s_it_i} \tag{39a}$$

$$\tilde{F}_{q,s_1\cdots,t_1\cdots} = \sum_{p',r_1'\cdots} D_{p'qr_1'r_2's_1s_2t_1}\bar{B}_{p'qr_3'r_2's_3s_2t_2} \prod_{i=4,6\cdots} B_{p'qr_{i-1}'r_i's_{i-1}s_it_{i-1}}\bar{B}_{p'qr_{i+1}'r_i's_{i+1}s_it_i} \tag{39b}$$

The error function $|\tilde{E} - \tilde{F}|^2$ is a quadratic in $D$, with coefficients given by products of $A$, $B$, $C$. This allows us to solve a linear matrix equation to find the optimal value of $D$ that minimizes $|\tilde{E} - \tilde{F}|^2$.

The expectation value is formally given by

$$\langle\hat{O}\rangle = \frac{\sum\langle\Omega|\hat{O}|\Psi\rangle}{\sum\langle\Omega|\Psi\rangle}. \tag{40}$$

The denominator corresponds to the usual path integral, where all boundary tensors are set to $T$. The numerator represents the path integral with the operator insertions. For finite systems, we continue the RG steps until only one or a few unit cells remain, at which point we can directly compute the path integral. For infinite systems, we compute the expectation value at each RG step for a few unit cells until convergence is reached.

For non-local operators or multiple local operators located at different sites, we need to introduce multiple impurity tensors. In essence, we combine impurity tensors after some number of RG steps, so the distance between them is controlled by the number of RG steps. In principle, the RG procedure can be tailored to accommodate any general set of insertions.

## B  Interpretation as a quantum circuit

Here we briefly mention an alternative perspective on the simplex tensor network state as a 2+1 dimensional Hermitian quantum circuit. We can group the $\Delta_3$s into local operators that act on an intersection plane. To illustrate this idea, we partition all vertices into three sets: $a$, $b$ and $c$, as shown in figure 12. There are three sets of operators $X_a$, $Y_b$ and $Z_c$, centered at different sets of sites $a$, $b$ and $c$, and act consecutively in the circuit as

$$\cdots \prod_{a'} X_{a'}^\dagger \prod_{b'} Y_{b'}^\dagger \prod_{c'} Z_{c'}^\dagger \prod_c Z_c \prod_b Y_b \prod_a X_a \cdots \tag{41}$$

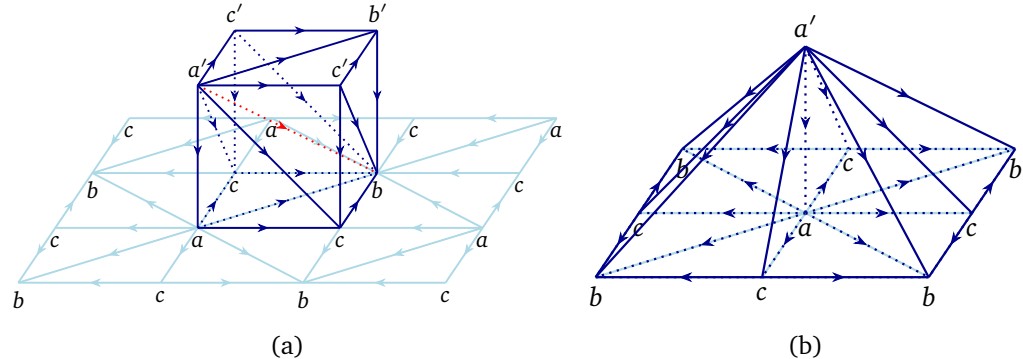

(a)                    (b)

Figure 12: **Operators of the quantum circuits.** In the left figure, the light blue lattice represents the 2d quantum states. The cubes are composed of the simplex tensors. The right figure shows how tensors centered at site $a$ construct an operator $X_a$ acting on the fields of $a$ and its surround edge and face indices, which in sketch means $X_a\Psi(\cdots, g_a, e_{ab_i}, e_{ac_i}, f_{ab_ic_i}, e_{ab_j}, \cdots) = \Psi'(\cdots, g_{a'}, e_{a'b_i}, e_{a'c_i}, f_{a'b_ic_i}, e_{a'b_j}, \cdots)$. The other two sets of operators are defined in a similar manner.

Since we define tensors of opposite chirality as being conjugate to one another, the product in equation 41 is Hermitian. Consequently, the RG flow can be interpreted as a coarse-graining procedure for this quantum circuit. At the TFP, the simplex tensors remain invariant under re-triangulation, which implies that the operators $X, Y, Z$ all commute and may form a stabilizer group. In the case of 2+1D $\mathbb{Z}_2$ gauge theory on a honeycomb lattice (which can be re-triangulated into ours), these operators correspond to the Hamiltonian terms in the toric code and double semion models, which is detailed in [34].

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
