# Peer review of "Simplex tensor network renormalization group for boundary theory of 3+1D symTFT"

_SciPost Physics_

## Round 1 · Referee Report · Anonymous (Referee 1) · 2025-4-25

Strengths

The idea of constructing gapless symmetric phases as boundary theories of symTFTs is highly interesting and timely, and pursuing this direction is clearly worthwhile.

The simplex tensor network representation of the boundary state is well-motivated and builds naturally on the earlier “strange correlator” formalism.

The renormalization group procedure is presented clearly, with helpful diagrams and a step-by-step explanation. The explicit examples in Sections 4.1 and 4.2 make the abstract construction more accessible and concrete for the reader.

Weaknesses

None.

Report

In this manuscript, the authors developed a symmetry-preserving renormalization group (RG) procedure to search for gapless boundary theories of symmetry topological field theories (symTFTs). The authors introduced a "simplex tensor network” and implemented a numerical RG algorithm by re-triangulations that preserve the symmetry, and apply this framework to a 3+1D Dijkgraaf-Witten (DW) theory with $Z_2$ gauge group, mapping out phase transitions between different boundary topological phases using both local and non-local order parameters.

The study of gapless symmetric phases and boundary theories of symTFTs is certainly a timely and important topic. The manuscript is well written and clearly structured. It should be a useful reference for readers working on symTFTs, gapless and topological phases of matter, and analytical/numerical aspects of tensor networks. In my opinion, it meets the acceptance criteria for SciPost Physics, and I recommend acceptance after some minor revisions.

Below, I list several comments and suggestions for the authors to consider.

i) There is a typo in page 6, the first line below the caption of Figure 2: “… is the wavefunction defined in (6), and $|\Psi(…)\rangle$ is some state defined also on …”. Here $|\Psi(…)\rangle$ should be $|\Omega(…)\rangle$.

ii) Regarding the choice of $|\Omega(…)\rangle$ mentioned in i), I also have two questions.

Are there any intuitions on the suitable choice of $|\Omega(…)\rangle$ for the resulting boundary theory to be critical?

For the 3+1D case, the partition function representing the gapless boundary may correspond to a 3D conformal field theory. Then, the gauge symmetry, which is encoded in the local tensor of $|\Omega(…)\rangle$, should be inherited in the partition function. Does this provide a constraint (other than the global symmetry) to the possible type of 3D CFT appearing at the boundary?

iii) For the example in Sec. 4.1, the authors mentioned that the resulting symTFT partition function is equivalent to a 3D classical Ising model with coupling J. If one uses the same construction but instead replace the gauge group by $Z_3$, one would probably obtain a symTFT partition function representing the 3D classical Potts model with $Z_3$ symmetry, which has a first-order phase transition as a function of the tuning parameter. Does it mean that a continuous phase transition is not guaranteed by construction?

iv) Page 15: For the interpolation between SPT$_0$ and SPT$_1$, the authors mentioned an undetermined CFT other than Ising at the tips of the “crescent” regions.

Have the authors attempted finite-size scaling analysis (e.g., scaling of order parameters) to confirm whether the transitions are continuous and associated with a critical theory? Could it also be a first-order transition?

v) Two minor typos:

Page 14: “We emphasis our numerical setups …” => “We emphasize our numerical setups …”

Page 18: “… ground work for generlization to any finite discrete …” => “… ground work for generalization to any finite discrete …”

Requested changes

Mentioned in the above report.

Recommendation

Ask for minor revision

---

## Round 1 · Referee Report · Anonymous (Referee 2) · 2025-5-11

Strengths

1- Clearly written review of the necessary background material 2- Practical application of recent ideas on generalised symmetries, a field which has been lacking in concrete applications

Weaknesses

1- Significant overlap with previous work by a subset of the same authors ([1], Phys. Rev. X 14, 041033) 2- Numerical results not very well described 3- Unclear whether models beyond the simplest case considered here are numerically tractable

Report

The authors use recent ideas on constructing symmetric lattice models as boundary theories of higher dimensional topological orders (topological holography/symTFT/strange correlators) to study RG flows of a (3+0)d partition function to its gapped fixed points. The global $\mathbb Z_2$ symmetry is imposed by the bulk topolgical order, and the RG scheme is designed in such a way that this global $\mathbb Z_2$ symmetry is preserved along the flow. Using this method, the authors perform numerical computations of the phase diagram for some given class of initial partition functions.

I think the results in this paper are interesting and open up a promising path towards a better understanding of gapped phases and phase transitions in 2d quantum lattice models. The application of the ideas underlying the symTFT (and its older precursors) to actual computational physics is very promising, and is a welcome addition to a field with mostly formal results. That being said, the paper is somewhat unclear in places, and I would appreciate some clarification on several points. Provided that the authors can address these points, I am happy to recommend this paper for publication in SciPost Physics.

Requested changes

1- Throughout the paper, the authors talk about the renomalisation of a (2+1)d theory as the boundary of a (3+1)d topological order. The actual object under consideration is the overlap between a (3+1)d topologically ordered state (obtained from the DW state sum construction) and some other state. The result is a number that is conventionally interpreted as the partition function of a statistical mechanics model in (3+0)d, where all space dimensions have been discretised. In the present work however, this partition function is stated to describe a model in (2+1)d, presumably as the Wick-rotated version of the discrete time evolution of a (2+1)d quantum state. It would be useful to have some comments on this distinction, and whether they affect the result in any meaninful way.

2- I find that the precise methodology to arrive at the numerical results is not very clearly explained. As I understand it, the plots in Figure 7 are obtained by taking a tensor T[x,y] as a starting point, and computing expectation value of some order operator using the method outlined in Appendix A.3. This means that the result shown is the expectation value of the order operator at the RG fixed point corresponding to the initial tensor T[x,y], not the expectation value of the actual theory obtained from T[x,y]. If this is indeed the case, I would strongly suggest adding some additional explanation on these matters.

3- Despite the focus in the introduction and motivation on devising a specific RG flow and its corresponding gapped fixed points, the paper does not contain any numerics that show the flow of the initial theories T[x,y] to their supposed fixed points beyond the fact that they reproduce the expected expectation value at the RG fixed point. Does this RG scheme actually converge to the fixed point states between which the theories T[x,y] are claimed to interpolate, i.e., do the actual tensors that build up the partition function converge to the fixed point tensors? I understand that these tensors are only defined up to gauge freedom, but there should still be a way to quantify how well the numerically obtained fixed point tensors approximate the exact RG fixed point.

4- It is unclear how the present method would compare to other 3D tensor network RG schemes that manifestly impose a global symmetry. I would invite the authors to consider their RG scheme for the case where the bulk topolgical order is trivial; i.e., there is no symmetry being imposed. If this reduces to an existing RG scheme, then I would expect that the case with a non-trivial bulk topological order simply corresponds to a symmetric version of this RG scheme.

5- For the topological model under consideration, i.e. a $\mathbb Z_2$ DW theory with a trivial 4-th cohomology, there are at least two inequivalent tensor network representations of the (3+1)d state, corresponding to two different choices of gapped boundary for this model. One is the smooth boundary, which naturally reproduces the representation used in this work. The other is the rough boundary, which provides a simpler tensor network representation with only edge degrees of freedom, of bond dimension 2. Locally, these two tensor network representations are indistinguishable, only differing in the states they represent on manifolds with nontrivial topology. I am curious whether there is any difference in using these two representations for the numerics in this paper. The same happens in (2+1)d, where one finds 2 inequivalent representations of the toric code ground state; one of bond dimension 2 (rough boundary), and one of bond dimension 4 (smooth boundary).

Recommendation

Ask for major revision

---

## Editorial Decision

awaiting_resubmission